# Use of the Species Sensitivity Distribution Approach to Derive Ecological Threshold of Toxicological Concern (eco-TTC) for Pesticides

**DOI:** 10.3390/ijerph182212078

**Published:** 2021-11-17

**Authors:** Cristiana Rizzi, Sara Villa, Alessandro Sergio Cuzzeri, Antonio Finizio

**Affiliations:** Department of Earth and Environmental Sciences DISAT, University of Milano-Bicocca, Piazza della Scienza 1, 20126 Milano, Italy; cristiana.rizzi@unimib.it (C.R.); a.cuzzeri@campus.unimib.it (A.S.C.); antonio.finizio@unimib.it (A.F.)

**Keywords:** species sensitivity distribution (SSD), pesticides, environmental risk assessment, aquatic communities, ecological threshold of toxicological concern

## Abstract

The species sensitivity distribution (SSD) calculates the hazardous concentration at which 5% of species (HC_5_) will be potentially affected. For many compounds, HC_5_ values are unavailable impeding the derivation of SSD curves. Through a detailed bibliographic survey, we selected HC_5_ values (from acute toxicity tests) for freshwater aquatic species and 129 pesticides. The statistical distribution and variability of the HC_5_ values within the chemical classes were evaluated. Insecticides are the most toxic compounds in the aquatic communities (HC_5_ = 1.4 × 10^−3^ µmol L^−1^), followed by herbicides (HC_5_ = 3.3 × 10^−2^ µmol L^−1^) and fungicides (HC_5_ = 7.8 µmol L^−1^). Subsequently, the specificity of the mode of action (MoA) of pesticides on freshwater aquatic communities was investigated by calculating the ratio between the estimated baseline toxicity for aquatic communities and the HC_5_ experimental values gathered from the literature. Moreover, we proposed and validated a scheme to derive the ecological thresholds of toxicological concern (eco-TTC) of pesticides for which data on their effects on aquatic communities are not available. We proposed eco-TTCs for different classes of insecticides, herbicides, and fungicides with a specific MoA, and three eco-TTCs for those chemicals with unavailable MoA. We consider the proposed approach and eco-TTC values useful for risk management purposes.

## 1. Introduction

Species may differ in their sensitivity towards a particular chemical [1,2,3], and this can be due to differences in behavioural, physiological, morphological, and life-history traits [4].

The recognition of these differences in sensitivity has led to the concept of species sensitivity distribution (SSD) being defined, which assumes that the sensitivities of a given number of species can be described by parametric or non-parametric statistical distribution functions [5,6,7,8,9,10].

The available ecotoxicological dataset (based on acute or chronic toxicity data) is considered representative of this distribution and is used to describe the SSD curve, from which it is possible to estimate the hazardous concentration for x% of the species (HCx). As the default threshold of significance is 5%, it is called HC_5_. For instance, if the SSD is derived from acute toxicity data (lethal concentration for 50% (LC_50_)), then HC_5_ is the concentration below the LC_50_ for 95% of the species [10].

To date, the SSD method is the noteworthy approach to determine a safe concentration for chemicals in terrestrial and aquatic ecosystems, and thus, the procedures of ecological risk assessment (ERA) make use of this approach. For instance, HC_5_ is divided by a chosen appropriate assessment factor (AF) to extrapolate the predicted no-effect concentration (PNEC), which is the most frequently used endpoint for summarising the overall hazard assessment in ERA [11,12,13]. The magnitude of variability in species’ sensitivity to toxicants is related to the chemical mode of action (MoA) [14,15,16,17,18,19].

Verhaar et al. [20] developed a scheme that enables the classification of chemicals based on their mode of toxic action into (a) inert chemicals (non-polar narcotics), (b) less inert chemicals (polar narcotics), (c) reactive chemicals, and (d) specifically acting chemicals. In principle, each organic compound can act as a non-polar narcotic, and consequently, this mode of action is considered the baseline toxicity. Quantitative structure-activity relationship (QSAR) equations have been proven successful in predicting the toxicity of non-polar narcotic compounds using a single attribute of the compounds (log Kow = the n-octanol-water partition coefficient) as a predictive descriptor [21]. According to Tremolada et al. [18], the distance from the baseline can be used as a measure of the specificity of the mode of action. The greater the distance is, the greater is the specificity of the mechanism of action. The measure of this distance can be calculated by means of the toxicity ratio (TR = LC_50_baseline/LC_50_experimental), where the LC_50_ baseline is obtained through the application of a QSAR equation and the LC_50_ experimental from laboratory tests.

The classification scheme proposed by Verhaar et al. [20] was based on acute toxicity data for a single species (the guppy (*Poecilia reticulata*)). Other MoA classification schemes were successively proposed by Kienzler et al. [22,23]. However, more recently, a study [24] proposed to expand the approach proposed by Verhaar et al. [20] at the hierarchical scale of aquatic communities. These authors proposed two different QSAR equations to predict the HC_5_ values for narcotic and polar narcotic compounds. In particular, significant correlations were found between HC_5_ and log Kow for several classes of non-polar narcotic compounds, indicating that hydrophobicity is still the key determinant of toxicity for aquatic communities.

By analogy with the approach proposed by Veith et al. [21] for single species, the obtained QSAR equation was then utilised to define the baseline toxicity for aquatic communities.

In the last few years, the concept of the ecological threshold of concern (eco-TTC) has been proposed as a pragmatic approach useful in screening-level ecological risk assessment [25,26,27]. Indeed, for chemicals with little or no ecotoxicological information, eco-TTC allows identifying the exposure threshold, below which there is a negligible risk for aquatic organisms. An eco-TTC is calculated starting from the probability distribution of PNECs of toxicologically or chemically similar groups of chemicals. In this, the eco-TTC differs from HC_5_ since, following the framework of PNEC calculation, it may combine acute and chronic toxicity data and consider the use of the AFs. On the other hand, as previously noted, HC_5_ values can be used to derive PNECs, which could be incorporated in the eco-TTC calculation [27].

In this framework, this study has the twofold aims of deepening the knowledge of the effects these substances have on freshwater communities by analysing the HC_5_ values (based on acute toxicity data) of a variety of pesticides gathered from the literature and proposing a scheme for deriving the ecological threshold of toxicological concern (eco-TTC) which holds great potential for risk management purposes.

## 2. Materials and Methods

### 2.1. Selection of Data

Approximately 240 HC_5_ values were collected from a wide variety of literature sources, with a total of 129 different pesticides once the data were processed. In particular, attention was focused on the HC_5_ values derived from the interpolation, consistent with the SSD curve model, of acute toxicity data (L(E)C_50_). The collection was conducted by consulting scientific journals, Google Scholar, websites of publishing houses (such as ScienceDirect, Elsevier, Wiley, and Springer), and CuriosOne (the library search service of the University of Milano-Bicocca).

The available literature was searched by running a simple search string consisting of specific keywords along with Boolean operators and a wildcard symbol (*). The search string was constructed as follows: (Pesticides* OR species sensitivity distribution* OR HC_5_*), occasionally including the names of specific compounds whenever more in-depth information was deemed necessary. Only verified peer-reviewed articles were included in this study. Each HC_5_ value was treated as an individual entry into the database. Details such as the year of publication and the number and typologies of species used to derive the SSD curves were also included in the database. The substances were then grouped according to their chemical class and pest target (i.e., insecticides: carbamates, organochlorines, organophosphorus compounds, pyrethroids, and neonicotinoids; herbicides: sulphonylureas, s-triazines, and phenylureas; fungicides: strobilurins and azoles). Finally, the HC5 values were all converted to molar basis (µmol L^−1^) using the molecular weight (MW) of the compound. Physical-chemical properties (MW and log Kow) of the studied compounds are reported in Appendix A.

The selection of data was made by considering the following aspects:Original studies reporting HC_5_ values were published in peer-reviewed journals;The HC_5_ values reported in the studies passed the test of normality (e.g., Anderson-Darling goodness-of-fit test);The HC_5_ values were referred to the same toxicological endpoints (e.g., growth, mortality).

Moreover, in 24 cases, more than one HC_5_ value was found in the literature. In these cases, the redundancy was accounted for by implementing a precautionary approach so that for every compound, the representative HC_5_ datum selected was the lowest one, to ensure a good level of protection for the most sensitive species in case of data bias.

Finally, we eliminated from the dataset those HC_5_ values which were extrapolated from SSD curves derived using less than 6 organisms. All the data are reported in Appendix A, together with the selection criteria.

We are aware that the collection of data from a literature survey may have introduced a certain degree of uncertainty in our results, particularly for the proper determination of eco-TTCs. Nevertheless, we highlight that most of the HC_5_ values were obtained from a few studies (≈85% of the selected HC_5_ were gathered from 6 studies), which indicates the homogeneity of data. Moreover, this study is not intended to accurately determine eco-TTC values but rather to propose a validated procedure for their evaluation.

### 2.2. Toxicity Ratios

As previously reported, the deviation from the “baseline” toxicity [20,21] can be assumed as a possible criterion to define the specificity of a mode of action for chemical compounds. This assumption is based on the consideration that the biological activity of “narcotic type” compounds lacks a specific mode of action and is completely dependent on the hydrophobicity, expressed as log Kow. Equation (1), which describes the QSAR equation proposed by Finizio et al. [24].
log 1/HC_5aqcom_ = 4.52 + 1.05 log K_ow_ (R^2^_adj_ = 0.97; *n* = 28; *p* < 0.0001)(1)
where HC5aqcom (µmol L^−1^) is the fifth percentile hazardous concentration for aquatic communities for non-polar narcotic compounds. 

A possible mode to measure the specificity of a mode of action is the use of the Toxicity ratio (TR) or the ratio between the estimated baseline toxicity (using QSAR equations) and the experimental toxicity data (i.e., LC_50_) (Equation (2)). In this study, we calculated the TR by estimating the baseline toxicity for the aquatic communities using Equation (1).
TR_aqcom_ = HC_5baseline aqcom)_/HC_5,exper._(2)
where

TR_aqcom_ = toxicity ratio for the aquatic communities;

HC_5 baseline aqcom_ = Hazardous Concentration 5% baseline toxicity for the aquatic communities calculated using Equation (1);

HC_5_,_exper_. = hazardous derived from experimental toxicity data (from the literature survey).

Based on the obtained logarithm of the toxicity ratio (log TR), pesticides were classified into the categories proposed by Tremolada et al. [18] and are reported in Table 1.

### 2.3. Ecological Threshold of Toxicological Concern (eco-TTC)

eco-TTC values were derived from the statistical analysis of the collected HC_5_ and by the application of an assessment factor (AF). We used Microsoft Excel to calculate statistical parameters, such as the mean, standard deviation, and 95th percentiles values (HC_5_median, HC_595th perc_, eco-TTC) for different categories and classes of pesticides. Briefly, in the first instance, we calculated the fifth percentile of all available HC_5_ for the three categories of pesticides (insecticides, herbicides, and fungicides). Subsequently, we clustered the substances based on their MoA. Finally, the calculated fifth percentile HC_5_ values were divided by an AF of 5, as suggested by the technical guidance documents [28] to derive the PNEC from the HC_5_ values (PNEC = HC_5_/5). In the absence of a known MoA (or when the MoA is not present in the scheme), we suggest using eco-TTCs of the three categories of pesticides.

## 3. Results

A synopsis of the collected data is reported in Table 2. More information about the selected HC_5_ values, the number, and typologies of tested organisms to derive the SSD curves can be found in Appendix A. From the bibliographic survey, we collected 146 HC_5_ values; however, due to the lack of robustness, those compounds for which the SSD were generated using <6 toxicity data were excluded from the successive analysis. Moreover, we excluded from the analysis also metabolites (i.e., p,p’-DDE, p,p’-DDD) which cannot be considered properly pesticides. Consequently, in the first part of the study, we report the results for 129 pesticides. All data were transformed and expressed with a molar basis (µmol L^−1^). In the following paragraphs (Section 3.1, Section 3.2 and Section 3.3), and for each category of pesticides, an overview of the HC_5_ values, together with statistical analysis, is reported.

### 3.1. Insecticides

In total, 48 HC_5_ values for insecticides were considered (Table 2 and Appendix A), mostly belonging to the classes of cyclodienes + cyclohexanes (CYC, *n* = 10), pyrethroids (PYR, *n* = 10), organophosphorus (OPs, *n* = 9), neonicotinoids (NEO, *n* = 9), and carbamates (CARBins, *n* = 4). HC_5_ values for these compounds were obtained from SSD curves, which were mainly generated using toxicity data obtained from tests on primary consumers (mainly arthropods), which are the most sensitive trophic level of aquatic communities for these compounds. The HC_5_ values for this category of pesticides span over five orders of magnitudes (HC_5min_ = 4.6 × 10^−6^ µmol L^−1^ and HC_5max_ = 6.2 × 10^−1^ µmol L^−1^ for cyfluthrin and tebufenozide, respectively) with a median value of 1.4 × 10^−3^ µmol L^−1^. However, this variability is highly reduced when a single class of compounds is considered, depending on the level of congenericity among compounds. PYRs are the most toxic class of compounds for the aquatic organisms with an HC_5 median_ of 1.8 × 10^−5^ µmol L^−1^ (±9.9 × 10^−5^) and an HC_5 95th perc_ of 5.3 × 10^−6^ µmol L^−1^. Other insecticides are in mean one to two orders of magnitude less toxic (OPs: HC_5 median_ = 1.1 × 10^−3^ µmol L^−1^ (±4.9 × 10^−4^); HC_5 95th perc_ = 1.6 × 10^−4^ µmol L^−1^; CARBins: HC_5 median_ = 7.3 × 10^−3^ µmol L^−1^; (±7.3 × 10^−3^); HC_5 95th perc_ = 8.7 × 10^−4^ µmol L^−1^; CYC: HC_5 median_ = 5.0 × 10^−3^ µmol L^−1^; (±1.1 × 10^−2^); HC_5 95th perc_ = 3.5 × 10^−4^ µmol L^−1^; NEO: HC_5 median_= 6.3 × 10^−3^ µmol L^−1^; (±1.0 × 10^−2^); HC_5 95th perc_ = 2.0 × 10^−3^ µmol L^−1^).

### 3.2. Herbicides

The HC_5_ values for 41 herbicides were retrieved from the literature (Table 2 and Appendix A). These values were derived from SSD curves obtained from toxicity data on primary producers. The HC_5_ values range from 1.9 × 10^−4^ µmol L^−1^ (HC_5min_ pentoxazone) to 1.0 × 10 10^2^ µmol L^−1^ (HC_5max_ pyriminobac–methyl). 

The most represented class of this category of pesticides are triazines (TR: HC_5 median_ = 3.5 × 10^−2^ µmol L^−1^ (±8.8 × 10^−2^) HC_5 95th perc_ = 7.9 × 10^−3^ µmol L^−1^), followed by ureas (UR: HC_5 median_ = 6.7 × 10^−2^ µmol L^−1^ (±1.); HC_5 95th perc_ = 9.8 × 10^−3^ µmol L^−1^) and sulphonylureas (SULF: HC_5 median_ = 5.9 × 10^−3^ µmol L^−1^ (±1.5 × 10^−2^); HC_5 95th perc_ = 6.9 × 10^−4^ µmol L^−1^). The latter seems to be the most hazardous class of herbicides for aquatic communities.

### 3.3. Fungicides

In Table 2 and Appendix A, the HC_5_ values for 40 fungicides are reported. The HC_5_ values for fungicides span over five orders of magnitudes (HC_5min_ = 4.2 × 10^−3^ µmol L^−1^ and HC_5max_ = 2.9 × 10^2^ µmol L^−1^ for thiram and hydroxyisoxazole, respectively), with a median value of 1.6 × 10^−1^. Azoles (AZ), carbamates (CARBfung), and strobilurines (STRO) are the fungicides classes most represented in the literature with HC_5 median_ values of 7.7 × 10^−1^ µmol L^−1^ (±1.0 × 10^2^), 7.1 × 10^−2^ µmol L^−1^ (±7.4 × 10^−2^), and 1.1 × 10^−1^ µmol L^−1^ (±3.7 × 10^−1^), respectively.

### 3.4. Effects of Pesticides on Aquatic Freshwater Communities in Relation to Their Specificity of Mode of Action (MoA)

The SSD curves represent the cumulative distribution of acute or chronic toxicity data for different species and are representative of the stress caused by chemicals to aquatic communities [10]. The basic postulation of the SSD approach is that sensitivities to a toxicant for a number of species follow a statistical distribution (preferably log-normal), which can be used in ecological risk assessment procedures as an effects assessment model. The probability of the data coming from a log-normal distribution must be tested by the goodness of fit (GOF) methods, such as the Kolmogorov-Smirnov or the Anderson-Darling tests (the last is recommended to datasets below 20) [28]. If the dataset fails the normality test, data must be rejected and cannot be used for SSD. One of the most frequent causes of failure of the test is due to the presence of species in the dataset which are very sensitive to the toxicant, while others are more tolerant, and this depends on the specificity of a mode of action [18,29]. Indeed, Finizio et al. [24] demonstrated that in absence of a specific mode of action (narcosis), the SSD curves can be derived using toxicity data from species, which are phylogenetically very different and cover the different trophic levels of the aquatic communities. Usually, the presence of a ‘break’ in the distribution between the sensitive and less sensitive species (due to the specific MoA of the toxicant) generates bimodal or multimodal SSD curves, causing the failure of the GOF test. To overcome this problem, a pragmatic choice is to split the taxa and derive the SSD curves from the subset of data that includes the most sensitive species. Thus, by protecting them automatically, also the less sensitive species will be kept safe. Pesticides are characterised by specific modes of action, and this explains why in our dataset the HC_5_ values present in the literature are mainly referred to the most sensitive species of the aquatic communities, particularly for herbicides (primary producers) and insecticides (mainly arthropods) (Appendix A). On the contrary, with a few exceptions, HC_5_ values for fungicides are derived from the SSD curves developed from toxicity data comprising species belonging to three trophic levels of aquatic ecosystems (AQCOM: primary producers, primary and secondary consumers), indicating the lesser specific mode of action of these substances (the target of fungicides are pathogens). Taking into consideration that the HC_5_ values present in the literature are derived from the SSD curves of the most sensitive species, we could conclude that insecticides are the most toxic compounds for aquatic communities (HC_5 median_ value = 1.4 × 10^−3^ µmol L^−1^), followed by herbicides (HC_5 median_ value = 3.3 × 10^−2^ µmol L^−1^) and fungicides (HC_5 median_ value = 7.8 µmol L^−1^). However, it should be noted that the selected HC_5_ values are referred only to acute toxicity data; consequently, no information about the chronic effects of these substances on aquatic communities can be derived. Moreover, ecological interactions between species (foodweb, competition) are not incorporated in SSD-based predictions and are not taken into consideration [30,31].

In the next paragraphs, we addressed two aspects: how the HC_5_ values present in the literature can be used to gain deeper insights into the effects of different categories and chemical classes of pesticides on freshwater aquatic communities, and how they can be used for risk management purposes.

In a pioneer work, Verhaar et al. [20] suggested the deviation from the ‘baseline toxicity’ as a possible benchmark to measure the specificity of the MoA of a compound for a single species. The assumption is that in the absence of a specific mechanism of action, the chemical behaves as a narcotic, and its toxicity is exclusively dependent on its hydrophobicity, measured by log Kow. The distance from the baseline toxicity can be assumed to indicate the MoA of the compounds. On this basis, the authors proposed a classification scheme in which each class is characterised by a specific mean TR (TR = LC_50 baseline_/LC_50 experimental_). The greater the TR, the greater the distance from the baseline toxicity, and the greater the specificity of the MoA. Thereafter, Tremolada et al. [18] proposed a modification of the Verhaar scheme for pesticides, as reported in Table 1.

Recently, Finizio et al. [24] developed a QSAR model to predict HC_5_ for non-polar narcotic compounds based on log Kow (see Equation (1)). The model was then used to define the baseline toxicity for aquatic communities (HC_5 baseline aqcom_). As in Verhaar’s study on single species, in this study, we used HC_5 baseline aqcom_ to calculate the TRs for pesticides to reveal information about the specificity of the MoA of these substances at a higher level of the ecological hierarchical scale, the community level. Data shown in Figure 1 are detailed reported in Appendix A. In Figure 1, the distances from the HC_5 baseline aqcom_ for different classes of insecticides (Figure 1A), herbicides (Figure 1B), and fungicides (Figure 1C) are depicted. In the same figure (Figure 1E,F), the normal distribution approximations of the log TR of the different classes of pesticides are also reported.

From Figure 1 and Table 1, the following considerations can be made:

For insecticides (Figure 1A,D), PYR appears to be the most acutely toxic class of insecticides for aquatic communities, with an HC_5_ median value of 1.8 × 10^−5^ µmol L^−1^. Indeed, all the other classes of investigated insecticides were two orders of magnitude less toxic (see Table 1). However, it is interesting to note that for these compounds, the distance from the baseline toxicity (Figure 1A) was not very high. In fact, based on the TRs (Figure 1D), these insecticides are classified as reactive chemicals (mean log TR = 2.4), together with CYC (mean log TR = 1.9) and are less specific than OPs, CARBins (mean log TRs = 3.6 and 3.8, respectively) and neonicotinoids (mean log TR = 6.2), respectively. The classification of PYR and CYC as reactive chemicals clearly indicates that log Kow plays a key role in explaining the toxicity of both classes of compounds. Interestingly, these insecticides show an analogous mechanism of action as they act on the efflux of ions in cell membranes, thereby disrupting electrical signalling in the nervous system. Indeed, the insecticidal properties of PYR alter the function of voltage-gated Na+ channels in insect neuronal membranes [32], and CYC inhibits Cl-flux into the nerve by binding to the picrotoxinin site in the γ-aminobutyric acid (GABA) chloride ionophore complex. It could be hypothesised that these substances interact with membranes due to their lipophilic behaviour. In fact, it has been demonstrated that insecticides that act on ion channels, through partitioning, cause changes in membrane fluidity [33,34,35,36].

As previously reported, organophosphorus compounds and carbamates have similar levels of toxicity and log TRs (and can be classified as specifically acting chemicals). It is interesting to note that, in this case, these compounds share the same mechanism of action. Indeed, their toxicity towards animals is attributed to their ability to inhibit acetylcholinesterase (AChE), which catalyses the hydrolysis of the neurotransmitting agent acetylcholine (ACh) [37].

Finally, NEO is less toxic than PYR, showing a toxicity level comparable with that of OPs and CARBins. NEO acts as an agonist of the nicotinic acetylcholine receptors (nAChRs) of insect acetylcholine receptors (nAChRs) [38], which are polypeptides that respond to the neurotransmitter acetylcholine. NEOs are classified as highly specifically acting chemicals, and the high value of log TR is due to the log Kow. Indeed, they show the lowest log Kow values when compared with the other classes of organic insecticides (Appendix A), which is consistent with their outstanding systemic herbicidal properties [39].

For herbicides (Figure 1B and E), the most toxic class of herbicides to primary producers of the aquatic communities is sulphonylureas (HC_5 median_ = 5.9 × 10^−3^ µmol L^−1^). Indeed, they are one order of magnitude more toxic than triazines and ureas (HC_5 median_ = 6.7 × 10^−2^ µmol L^−1^ and 3.5 × 10^−2^ µmol L^−1^, respectively). Sulfonylureas inhibit acetolactate synthase (acetohydroxyacid synthase), thereby blocking the biosynthesis of the branched-chain amino acids leucine, isoleucine, and valine [40]. This inhibition leads to the rapid cessation of plant cell division and growth. Among herbicides, sulphonylureas also show the highest log TR (mean log TR = 5.0) and are classified as highly specifically acting chemicals; owing to the very low log Kow values of these compounds, which are analogous to NEO, they are systemic pesticides. UR and TR fall in the category of specifically acting chemicals with a very similar mean log TR of 2.75 and 3.11, respectively. Interestingly, both classes share the same mechanism of action (inhibition of photosynthetic electron transport).

As regards fungicides (Figure 1C,F), STROs (HC_5 median_ = 7.1 × 10^−2^ µmol L^−1^) are the most toxic class of the investigated fungicides and fall in the category of specifically acting chemicals (log TR = 2.2). Indeed, their MoA is quite specific, inhibiting mitochondrial respiration in fungi by interfering with the function of the cytochrome bc1 complex [41].

CARBfung (HC_5 median_ = 1.1 × 10^−1^ µmol L^−1^) contain a benzimidazole group and are classified as highly specifically acting chemicals (log TR = 4.5). The breakdown product of CARBfung, ethylene-bis-isothiocyanate sulphide, reacts with and inactivates the sulphhydryl groups of amino acids and enzymes within fungal cells, resulting in the disruption of lipid metabolism, respiration, and production of ATP [42]. Finally, AZOs are the least toxic compound among the investigated fungicides (HC_5 median_ = 7.7 × 10^−1^ µmol L^−1^) and are classified as reactive chemicals (mean log TR = 1.7). These compounds are steroid demethylation inhibitors, particularly for sterol 14α-demethylase. Although 14α-demethylase is present in a wide variety of organisms, it has been studied primarily in the context of fungi, where it plays an essential role in mediating membrane permeability [43].

Finally, it is not necessarily true that pesticides classified as highly specifically acting chemicals are more toxic than those belonging to other classes. For instance, NEO (highly specifically acting chemicals) are less toxic than PYR (reactive chemicals), and CARBfung are less toxic than STRO. This highlights the importance of absorption, distribution, metabolism, and excretion (ADME) in the explanation of toxic effects.

### 3.5. Use of Available HC5 and Mechanism of Action for Risk Assessment and Management

The introduction of the concept of ecological thresholds of toxicological concern (eco-TTC) has resulted from the upgrading of risk assessment procedures in recent years, the aim of which is to establish an exposure level for chemicals beneath which no significant risk to ecosystems is likely [26,27,44,45]. The use of eco-TTC reduces the need for ecotoxicological studies of untested chemicals in cases where it can be demonstrated that the expected exposure is unlikely to exceed a level of concern. Indeed, eco-TTC have been established and attributed to classes or groups of chemicals sharing the same MoA [46,47,48]. For instance, De Wolf et al. [46] derived the exposure thresholds of no concern for the aquatic compartment based on the Verhaar [20] categorisation scheme. Based on a detailed analysis of aquatic toxicity databases, the authors derived eco-TTCs for organic narcotic, polar narcotics, and reactive chemicals (MoA1, MoA2, and MoA3 according to the Verhaar scheme). However, specifically acting chemicals (MoA4), such as pesticides, were not considered in the analysis, and no eco-TTCs were derived for these compounds. In this paper, we propose a tree diagram scheme (Figure 2) which can be applied to define eco-TTCs for classes or groups of pesticides sharing the same MoA. The diagram is based on the worst-case approach and the classification proposed by Tremolada et al. [18] (Table 1). To validate the proposed scheme, we performed another bibliographic search to collect other HC_5_ values for pesticides which were not considered in the previous analysis. Successively, their relative eco-TTCs were calculated and compared with those proposed in Figure 2 for untested chemicals (Table 3). The analysis of the results clearly highlights the high capability of the proposed scheme to assign appropriate eco-TTC values to untested chemicals. Indeed, in almost all the analysed cases, the calculated eco-TTCs and those attributed to the class of compounds, showing the same mechanism of action are very similar. Moreover, the calculated eco-TTC shows higher toxicity values, which indicates that the worst-case assumption is respected.

## 4. Conclusions

Information on the toxicity of pesticides on freshwater aquatic organisms is fundamental to managing the risks posed by these contaminants and to regulatory standards aiming to preserve the structure and functionality of these ecosystems. To date, there are a large number of data aiming at evaluating the effects of these substances at the aquatic community level, and these data are generally derived from SSD studies and the derivation of HC_5_ values. However, these data are scattered in the literature, and this prevents a deepening view of their effects on aquatic ecosystems and a comparison among the different pesticides categories (herbicides, fungicides, and insecticides) and mechanisms of action. In this regard, it would be very useful to create new or implement existing databases to collect raw data and HC_5_ values. In this study, through a bibliographic search, we collected 129 HC_5_ values. These data were organised to evaluate the statistical distribution and variability of the HC_5_ values (all derived from SSD curves based on acute toxicity data) for categories and chemical classes. The results clearly indicated that insecticides are the most toxic category of pesticides for freshwater ecosystems (particularly PYR and CARBins), followed by herbicides (particularly SUL) and fungicides (CARBfung). The TRs were calculated to determine the specificity of the MoA of the pesticide classes. Finally, we proposed and validated a scheme to derive the eco-TTC for pesticides for which data on their effects on aquatic communities would be extremely beneficial for risk management purposes.

## Figures and Tables

**Figure 1 ijerph-18-12078-f001:**
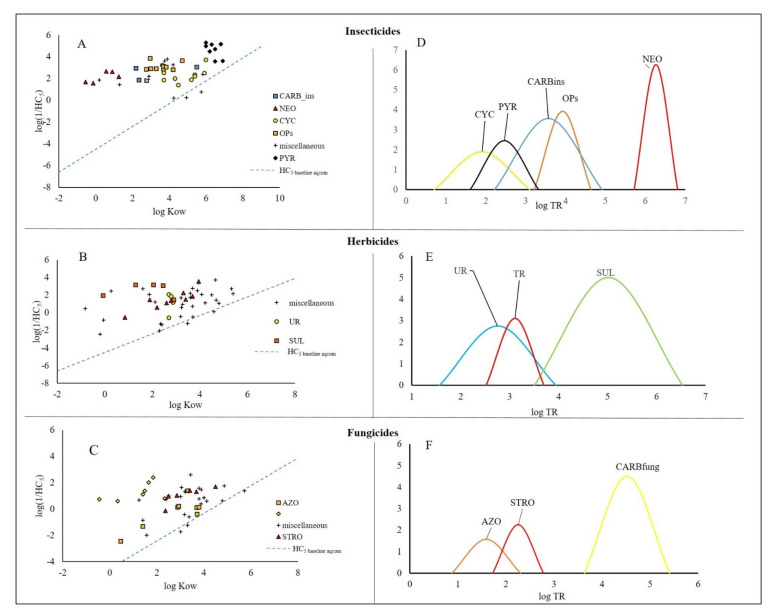
Distance from the baseline toxicity for aquatic communities (HC_5 baseline aqcom_ = dashed line calculated according to Equation (1)) and log TR distribution of different classes of pesticides: insecticides (**A**,**D**); herbicides (**B**,**E**); fungicides (**C**,**F**).

**Figure 2 ijerph-18-12078-f002:**
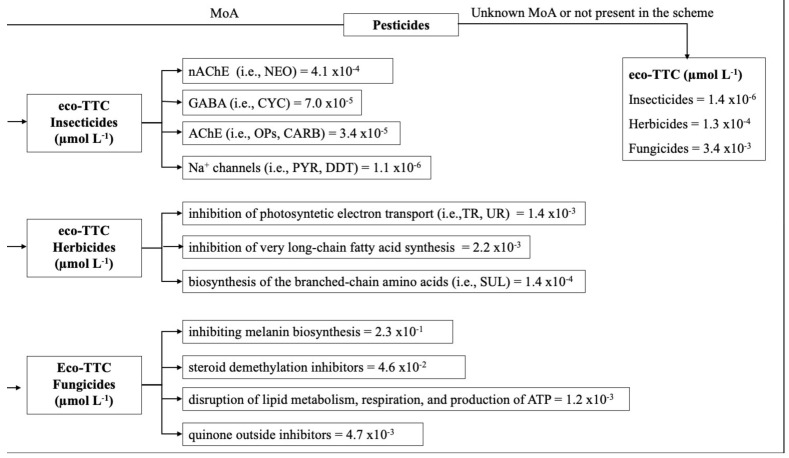
Tree diagram scheme to define eco-TTCs expressed in μmol L^−1^ for classes or groups of pesticides sharing the same mode of action.

**Table 1 ijerph-18-12078-t001:** Classification scheme of pesticides based on mechanisms of action [18].

Class	Mechanism of Action	Log TR
I	Non-polar narcotics	−0.5 < log TR < 0.5
II	Polar narcotics	0.5 < log TR < 1.5
III	Reactive chemicals	1.5 < log TR < 2.5
IV	Specifically acting chemicals	2.5 < log TR < 4
V	Highly specifically acting chemicals	log TR > 4

**Table 2 ijerph-18-12078-t002:** Synopsis of the median HC_5_ values (µmol L^−1^) with their range of variability (minimum, maximum, and 95th percentile values for all the investigated HC_5_) for classes of pesticides.

Categories	Chemical Class	Nr of Comp.	HC_5min_ (µmol L^−1^)	HC_5max_ (µmol L^−1^)	HC_5median_ (µmol L^−1^)	HC_5 95th perc_ (µmol L^−1^)
insecticides	-	48	4.6 × 10^−6^ (cyfluthrin)	6.2 × 10^−1^ (tebufenozide)	1.4 × 10^−3^	6.9 × 10^−6^
	CARBins	4	8.4 × 10^−4^ (carbosulphan)	1.4 × 10^−2^ (fenobucarb)	7.2 × 10^−3^	8.7 × 10^−4^
	NEO	5	2.0 × 10^−3^ (imidacloprid)	2.4 × 10^−2^ (thiamethoxam)	6.3 × 10^−3^	2.0 × 10^−3^
	CYC	10	1.8 × 10^−4^ (endrin)	3.7 × 10^−2^ (chlordecone)	5.0 × 10^−3^	3.5 × 10^−4^
	OPs	9	9.7 × 10^−5^ (chlorpyrifos)	1.5 × 10^−3^ (fenthion)	1.1 × 10^−3^	1.6 × 10^−4^
	PYR	9	4.6 × 10^−6^ (cyfluthrin)	2.4 × 10^−4^ (permethrin)	1.8 × 10^−5^	5.3 × 10^−6^
	miscellaneus	11	-	-	-	-
herbicides	-	41	1.9 × 10^−4^ (pentoxazone)	1.0 × 10^2^ (pyriminobac-CH3)	3.3 × 10^−2^	6.6 × 10^−4^
	TR	7	5.4 × 10^−3^ (prometryn)	2.6 × 10^−1^ (simazine)	3.5 × 10^−2^	7.9 × 10^−3^
	SUL	4	6.6 × 10^−4^ (cyclosulphamuron)	3.3 × 10^−2^ (propyrisulphuron)	5.9 × 10^−3^	6.9 × 10^−4^
	UR	5	8.1 × 10^−3^ (diuron)	3.7 (daimuron)	6.7 × 10^−2^	9.8 × 10^−3^
	miscellaneous	25	-	-	-	-
fungicides	-	40	4.2 × 10^−3^ (thiram)	2.9 × 10^2^ (hydroxyisoxazole)	1.6 × 10^−1^	1.7 × 10^−2^
	AZ	8	4.2 × 10^−2^ (epoxiconazole)	2.9 × 10^2^ (hydroxyisoxazole)	7.7 × 10^−1^	2.3 × 10^−1^
	CARBfung	7	4.2 × 10^−3^ (thiram)	1.8 × 10^−1^ (maneb)	1.1 × 10^−1^	5.9 × 10^−3^
	STRO	5	2.0 × 10^−2^ (trifloxystrobin)	1.0 × 10^−1^ (azoxystrobin)	7.1 × 10^−2^	2.3 × 10^−2^
	miscellaneous	20	-	-	-	-

**Table 3 ijerph-18-12078-t003:** Comparison of the calculated eco-TTC and literature data of HC_5_ (expressed in μg L^−1^ and μmol L^−1^) and the eco-TTC assigned to classes of pesticides exhibiting the same mechanism of action according to this procedure scheme (eco-TTC values calculated using an AF = 5).

Mechanism of Action	Chemical	HC_5_ (µg L^−1^)	Reference	HC_5_ (µmol L^−1^)	eco-TTC Calculated	eco-TTC (Mechanisms of Action Class)
AChE inhibitors	Insecticides					
	chlorfenvinphos	1.1 × 10^−1^	[49]	3.1 × 10^−4^	6.2 × 10^−5^	3.4 × 10^−5^
	dipterex	2.2 × 10^−1^	[49]	7.6 × 10^−4^	1.5 × 10^−4^	
	ethoprophos (nematocide)	3.1	[50]	1.3 × 10^−2^	2.6 × 10^−3^	
	fenamiphos	8.2 × 10^−1^	[50]	2.7 × 10^−3^	5.4 × 10^−4^	
	terbufos	1.0 × 10^−1^	[50]	3.5 × 10^−4^	6.9 × 10^−5^	
	Herbicides					
Inhibition of very long-chain fatty acid synthesis	acetochlor	1.1 × 10^1^	[50]	4.1 × 10^−2^	2.0 × 10^−3^	2.2 × 10^−3^
	alachlor	2.7	[51]	1.0 × 10^−2^	3.0 × 10^−2^	
Inhibition of very long-chain fatty acid synthesis	chlorotoluron	3.2 × 10^1^	[52]	1.5 × 10^−1^	8.2 × 10^−3^	
	ametryn	2.3 × 10^−1^	[50]	1.0 × 10^−3^	2.0 × 10^−4^	
	Fungicides					
Steroid demethylation inhibitors	difeconazole	1.0 × 10 ^2^	[50]	2.5 × 10^−1^	5.0 × 10^−2^	4.6 × 10^−2^

## Data Availability

Data is contained within the article or Appendix A.

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
