# Peer review of "Use of the Species Sensitivity Distribution Approach to Derive Ecological Threshold of Toxicological Concern (eco-TTC) for Pesticides"

_ijerph, 2021, doi:10.3390/ijerph182212078_

Round 1

Reviewer 1 Report

In the study, through a bibliographic search, were collected 129 HC5 values. These data were organized to evaluate the statistical distribution and variability of the HC5 values (all derived from SSD curves based on acute toxicity data) for categories and chemical classes. The results clearly indicated that insecticides are the most toxic category of pesticides for freshwater ecosystems (particularly PYR and Carbins), followed by herbicides (particularly SUL) and fungicides (Carbfung). The TRs was calculated to determine the specificity of the MoA of the pesticide classes. Finally, we proposed and validated a scheme to derive the eco-TTC for pesticides for which data on their effects on aquatic communities would be extremely beneficial for risk management purposes.

This is very interesting review paper and I propose to publish it after minor revisions as below.

General Remarks

Abstract

  • Please do not use abbreviation in abstract or provide the full name.

 Detailed remarks

 Line 157 – …”Eco-TTC values were derived from the statistical analysis of the collected HC”… Methods of statistical analysis should be added

  1. Table 2, 3 should be self-explaining. Please add proper information.
  2. Figure 1, 2 should be self-explaining. Please add proper information.
  3. 50% of the cited literature is from before 2010, including 26% from before 2000. Please comment.

Author Response

Rev. 1

In the study, through a bibliographic search, were collected 129 HC5 values. These data were organized to evaluate the statistical distribution and variability of the HC5 values (all derived from SSD curves based on acute toxicity data) for categories and chemical classes. The results clearly indicated that insecticides are the most toxic category of pesticides for freshwater ecosystems (particularly PYR and Carbins), followed by herbicides (particularly SUL) and fungicides (Carbfung). The TRs was calculated to determine the specificity of the MoA of the pesticide classes. Finally, we proposed and validated a scheme to derive the eco-TTC for pesticides for which data on their effects on aquatic communities would be extremely beneficial for risk management purposes.

This is very interesting review paper and I propose to publish it after minor revisions as below.

General Remarks

Abstract

  • Please do not use abbreviation in abstract or provide the full name.

Accepted. Thank you for the suggestion

 Detailed remarks

 Line 157 – …”Eco-TTC values were derived from the statistical analysis of the collected HC”… Methods of statistical analysis should be added

Accepted. We have added the following sentence: We used Microsoft Excel to calculate statistical parameters, such as the mean, standard deviation and 95th percentiles values (HC5median, HC595th perc, eco-TTC) for different categories and classes of pesticides.

  1. Table 2, 3 should be self-explaining. Please add proper information.

Accepted. Thank you for the suggestion

  1. Figure 1, 2 should be self-explaining. Please add proper information.

Accepted. Thank you for the suggestion

  1. 50% of the cited literature is from before 2010, including 26% from before 2000. Please comment.

We have not added any comments regarding the publishing data collected in the literature. The oldest works refer to the definition of the mechanisms of action of substances used for the creation and subsequent validation of the proposed approach. Precisely because of the type of data sought (Mode of action) we believe that this information is not subject to changes over time.

Reviewer 2 Report

This study analyzed the hazardous concentration at which 5% of species will be potentially affected (HC5) values of a variety of pesticides based on acute toxicity data from literature. They wanted to propose a scheme for deriving the ecological threshold of toxicological concern which holds great potential for risk management purposes.

The article was well done. I have no comment on the structure. However, if they offered more information about their literature searching, it would be better for readers.

  1. They stated using search string consisting of specific keywords along with Boolean operation, but there was no specific database, such as Medline or Toxline?
  2. How many papers or reports were found initially? Inclusion criteria? Exclusion criteria?
  3. Please present the numbers used in Figure 1.

Author Response

This study analyzed the hazardous concentration at which 5% of species will be potentially affected (HC5) values of a variety of pesticides based on acute toxicity data from literature. They wanted to propose a scheme for deriving the ecological threshold of toxicological concern which holds great potential for risk management purposes.

The article was well done. I have no comment on the structure. However, if they offered more information about their literature searching, it would be better for readers.

  1. They stated using search string consisting of specific keywords along with Boolean operation, but there was no specific database, such as Medline or Toxline?

To the best of our knowledge, there are not any public database reporting the HC5s (Hazardous concentration of 5% per the species of a given community). The databases indicated by the reviewer report toxicity data for single species.

  1. How many papers or reports were found initially? Inclusion criteria? Exclusion criteria? And How many papers and reports were found initially? Criteria?

Of all the papers and reports initially found in the existing literature, about 20 made it to the “final selection”. During a primary exclusion process, some articles were ruled out because, for example, the way they expressed their results was incompatible with the other studies taken into consideration (e.g. HC5 expressed as toxic equivalents for a specific species), or because we could not find or assign a logKow to the chemical substances (such was the case for some nano materials like Zn and Cu oxides, oils and oil dispersant products). Later on, we included in our dataset only those data coming from a solid statistical analysis, and proceeded ruling out the studies with a strong bias in the experimental conditions (e.g. too specific water matrixes), unclear or unobtainable references, and outliers for substances we had no comparison for (e.g. Glyphosate). 

  1. Please present the numbers used in Figure 1.

Agreed. We have added a sentence in the main text quoting the table, in the Supporting information, where all the raw data is reported.

Reviewer 3 Report

I have reviewed the manuscript entitled "Use of the Species Sensitivity Distribution Approach to Derive Ecological Threshold of Toxicological Concern (eco-TTC) for Pesticides" by Rizzi et al. for publication in International Journal of Environmental Research and Public Health. The authors selected the HC5 values for freshwater aquatic species and 129 pesticides from literature data, and based on these data, they proposed a scheme for deriving the ecological threshold of toxicological concern which holds great potential for risk management purposes. The work is well organized, and the subject matter is very interesting, so I recommend it for publication after minor revision.

Comments:

1) Check in the document that the format of the journal is respected (for example the text must be justified).

2) Since the study involved the toxicology of pesticides, it must be taken into account that some of these compounds are considered endocrine disrupting compounds. I suggest improving the introduction section by adding some aspects regarding risk assessment for these compounds (non‐monotonic dose‐response relationships and low‐dose effects) that are highly debated in literature (see Pironti, C.; Ricciardi, M.; Proto, A.; Bianco, P.M.; Montano, L.; Motta, O. Endocrine-Disrupting Compounds: An Overview on Their Occurrence in the Aquatic Environment and Human Exposure. Water 2021, 13, 1347. https://doi.org/10.3390/w13101347).

3) I suggest moving equation 1 in the 2.2 section to make all the equations clearer, because lines 141-146 are confusing.

4) Since the manuscript is organized by separating the results from their discussion, it would probably be more appropriate to also report Figure 1, Table 3, and Figure 2 in the results section in order to make the text in the discussion section more fluent.

5) I suggest increasing the size of the writings in figure 2.

Author Response

  • Check in the document that the format of the journal is respected (for example the text must be justified).

Accepted. Thank you for the suggestion.

  • Since the study involved the toxicology of pesticides, it must be taken into account that some of these compounds are considered endocrine disrupting compounds. I suggest improving the introduction section by adding some aspects regarding risk assessment for these compounds (non‐monotonic dose‐response relationships and low‐dose effects) that are highly debated in literature (see Pironti, C.; Ricciardi, M.; Proto, A.; Bianco, P.M.; Montano, L.; Motta, O. Endocrine-Disrupting Compounds: An Overview on Their Occurrence in the Aquatic Environment and Human Exposure. Water2021, 13, 1347. https://doi.org/10.3390/w13101347).

Not accepted. The paper quoted by the reviewer relates to the presence in the environment of pesticides whit endocrine-disrupting properties and the effects these chemicals induce in humans. We believe that this, even is it a very important aspect correlated to the presence of some pesticide in the environment, is out of the scope of this paper.

  • I suggest moving equation 1 in the 2.2 section to make all the equations clearer, because lines 141-146 are confusing.

Accepted. Thank you for the suggestion.

4) Since the manuscript is organized by separating the results from their discussion, it would probably be more appropriate to also report Figure 1, Table 3, and Figure 2 in the results section in order to make the text in the discussion section more fluent.

Accepted. Thank you for the suggestion.

5) I suggest increasing the size of the writings in figure 2.

Accepted. Thank you for the suggestion.
